# Healthcare Management of Human African Trypanosomiasis Cases in the Eastern, Muchinga and Lusaka Provinces of Zambia

**DOI:** 10.3390/tropicalmed7100270

**Published:** 2022-09-27

**Authors:** Allan Mayaba Mwiinde, Martin Simuunza, Boniface Namangala, Chitalu Miriam Chama-Chiliba, Noreen Machila, Neil E. Anderson, Peter M. Atkinson, Susan C. Welburn

**Affiliations:** 1Department of Public Health, Mazabuka Municipal Council, Mazabuka P.O. Box 620033, Zambia; 2School of Veterinary Medicine, University of Zambia, Lusaka P.O. Box 50110, Zambia; 3Institute of Economic and Social Research (INESOR), University of Zambia, Lusaka P.O. Box 30900, Zambia; 4School of Biomedical Sciences, Edinburgh Medical School, College of Medicine and Veterinary Medicine, University of Edinburgh, Edinburgh EH4 2XU, UK; 5The Royal (Dick) School of Veterinary Studies and the Roslin Institute, University of Edinburgh, 49 Little France Crescent, Edinburgh EH25 9RG, UK; 6Centre for Health Informatics, Computing and Statistics (CHICAS), Lancaster Medical School, Faculty of Health and Medicine, Lancaster University, Lancaster LA1 4YQ, UK; 7Zhejiang University-University of Edinburgh Joint Institute, Zhejiang University, International Campus, 718 East Haizhou Road, Haining 314400, China

**Keywords:** human African trypanosomiasis, sleeping sickness, *T. b. rhodesiense*, health care, spatial distribution, Zambia

## Abstract

Human African trypanosomiasis (HAT) is a neglected tropical disease that has not received much attention in Zambia and most of the countries in which it occurs. In this study, we assessed the adequacy of the healthcare delivery system in diagnosis and management of rHAT cases, the environmental factors associated with transmission, the population at risk and the geographical location of rHAT cases. Structured questionnaires, focus group discussions and key informant interviews were conducted among the affected communities and health workers. The study identified 64 cases of rHAT, of which 26 were identified through active surveillance and 38 through passive surveillance. We identified a significant association between knowledge of the vector for rHAT and knowledge of rHAT transmission (*p* < 0.028). In all four districts, late or poor diagnosis occurred due to a lack of qualified laboratory technicians and diagnostic equipment. This study reveals that the current Zambian healthcare system is not able to adequately handle rHAT cases. Targeted policies to improve staff training in rHAT disease detection and management are needed to ensure that sustainable elimination of this public health problem is achieved in line with global targets.

## 1. Introduction

Human African trypanosomiasis (HAT) is a re-emerging public health problem caused by two protozoan parasitic infections, *Trypanosoma brucei rhodesiense* and *Trypanosoma brucei gambiense*, with epidemics being recorded in rural parts of Africa [1]. These two extracellular hemoflagellate protozoan parasites are transmitted by insect vectors of the genus *Glossina* (tsetse flies) [2]. In the Rhodesiense (rHAT) form, the transmission of the parasite by tsetse flies leads to occasional infection of humans, whereas in the *T. b. gambiense* (gHAT) form, humans are the main reservoir in the transmission cycle of the disease [3]. Thus, the two forms of the disease caused by rHAT and gHAT parasites are characterised by different epidemiological and clinical patterns and approaches to patient management [4]. *T. b gambiense* accounts for 98% of reported HAT cases. During the early stage of the disease, occurring several months to years after the initial bite by an infected fly, the parasites are confined to the blood and lymphatic circulation systems, whereas invasion of the central nervous system defines the late or chronic stage of gHAT [5,6]. *Trypanosoma brucei rhodesiense* accounts for 2% of reported cases. Unlike gHAT, rHAT is relatively more acute, early-stage symptoms observed within weeks or months after a bite from an infected fly. Furthermore, rHAT progresses rapidly to invade the central nervous system [5,6].

The global target is to eliminate HAT as a public health problem [7]. However, achieving this milestone requires affected countries to have a strong health system in place for early identification of clinical signs and symptoms of HAT, case referral, laboratory diagnosis and effective treatment instituted sufficiently early to minimise serious drug reactions and mortality [8]. New or improved technologies are also needed to tackle this disease, which often affects the poorest populations in developing countries [9]. Notably, new tools have been developed for gHAT, although few advancements have been made for rHAT.

Zambia has 10 provinces, which are divided into 105 districts [10]. A study by Mwanakasale and Songolo on the spatial distribution of rHAT in seven districts of Zambia revealed that no new rHAT cases were reported in these districts after January 2000 [11]. Another study covering 26 districts in Zambia used retrospective hospital records to show that three districts reported rHAT cases consistently between 2005 and 2014; one district had cases between 2000 and 2004 only; and three districts appeared to support emerging or re-emerging rHAT transmission, indicating that Zambia still had areas with active rHAT transmission [12]. In a recent study, we reported the occurrence of rHAT cases in Mpika and Chama Districts (Muchinga Province), Mambwe District (Eastern Province) and Rufunsa District (Lusaka Province) [13]. Recent studies have also reported rHAT in Kafue National Park 50 years since the last case of the disease [14]. Despite efforts by the World Health Organisation (WHO), for example, by promoting public–private partnerships to move toward the elimination of rHAT as a public health problem [15], serious challenges exist in Zambia with respect to the effective control of the disease [7], such as poor surveillance and weak local medical services [13].

The aims of this study were to identify gaps in the adequacy of the healthcare delivery system in relation to the control and management of rHAT in the endemic foci of the disease in Zambia, to assess the population at-risk and to map the distribution of current rHAT cases in the Chama, Rufunsa, Mpika and Mambwe Districts of Zambia. We also examined the environmental factors associated with the transmission of rHAT in relation to the geographical location of cases and disease transmission. These aspects are not highlighted adequately in the available literature on Zambia. In a previous study [13], we reported on the economic and social consequences of rHAT in Zambia, considering disability-adjusted life years (DALYs); healthcare cost associated with admission of an rHAT patient; and social matters, such as stigma [13]. In this study, we examined the healthcare delivery system as it relates to the management and diagnosis of rHAT cases. 

## 2. Materials and Methods

### 2.1. Ethical Permissions and Consent to Participate

Research and ethical clearance was obtained from and granted by the University of Zambia Biomedical Research Ethics Committee (reference number 011-09-13). Written consent, including willingness to participate in the study, was obtained from each subject enrolled in the study. All participants were guaranteed confidentiality of their personal information at the individual level.

### 2.2. Study Site and Design

In this study, we used the same dataset as in our previous study [13], which is available from the Natural Environment Research Council (NERC) Environmental Information Data Centre [16]. The study was undertaken in four districts of Zambia where sporadic cases of rHAT have been reported in the last decade, namely Mambwe in Eastern Province, Mpika and Chama in Muchinga Province and Rufunsa in Lusaka Province. 

The three study provinces, Eastern, Muchinga and Lusaka, have a tropical climate with three distinct seasons per year, known as the hot–dry season, the rainy season and the cold–dry season [17]. Muchinga Province has the highest altitude of 1393 m above sea level, with an average annual rainfall of 1110 mm and an average temperature ranging from 10.1 °C to 30 °C, whereas Eastern Province has the lowest altitude of 1045 m, with annual rainfall averaging 1000 mm and an average temperature ranging from 12.3 °C to 32.6 °C. Lusaka Province has an altitude of 1272 m, with annual rainfall of 800–1000 mm and an average temperature ranging from 10.1 °C to 31.6 °C [17,18]. A detailed description of the study area, including sources of livelihoods of the communities, is available elsewhere [13]. A cross-sectional and retrospective survey was conducted in the study areas to characterise the healthcare delivery system and map the spatial distribution of rHAT cases. The study involved data collection in health facilities that recorded cases of rHAT, as determined previously [12]. 

### 2.3. Methods for Data Collection

We used passive and active surveillance systems to obtain the data used in the present study. Passive surveillance consisted of infected people self-presenting to fixed health facilities with standard reporting of disease information while receiving treatment. Active surveillance involved researchers or healthcare providers actively searching for rHAT cases and related information from communities or health institutions between 2013 and 2014 through carefully designed and approved study protocols [16].

To determine the characteristics of rHAT in previously and currently infected rHAT patients, we used data from hospital registers and/or surveyed health practitioners and the community. The study included all four districts (Rufunsa, Chama, Mambwe and Mpika) with active cases, their health centres and patients diagnosed with rHAT from 2004 to 2014. Only first-level district first hospitals (*n* = 4) and their health centres (*n* = 7) that recorded rHAT cases during the period under review were selected in the study districts. The University Teaching Hospital was purposively selected, as it is the final referral centre for all hospitals in Zambia for the management of rHAT. Separate questionnaires were used for patients and health workers. Pretesting of the questionnaires was conducted to identify and rectify problems that might arise for both respondents (care giver) and interviewers about question content. The questionnaire was developed and administered by the lead author between September and November 2014.

From 2004 to 2012, patients were identified through passive surveillance during routine hospital visits. Patients accessing healthcare clinics with symptoms and suspected of rHAT were sent to the laboratory for rHAT confirmation through diagnosis by microscopy (examination of thin blood smears stained with Giemsa), and no further follow-up for diagnostic testing was undertaken. However, from 2013 to 2014, passively diagnosed rHAT patients were confirmed as having rHAT using microscopy of Giemsa-stained buffy coat smears, polymerase chain reaction (PCR) [19] and/or loop-mediated isothermal amplification (LAMP) [20] and subsequently followed-up more closely. Patient data were obtained through health centre or hospital records, and national statistics were obtained from the Ministry of Health. 

Structured questionnaires were administered to former and current rHAT patients or close relatives (care givers) in the case of a deceased person, as reported in [13]. The structured questionnaires aimed to assess the knowledge and practices of transmission among rHAT victims. The respondents were asked questions such as, “Do you understand the vector responsible for transmission?” and “…the mode of rHAT transmission?”. Furthermore, focus group discussions were conducted with the affected communities to provide an in-depth understanding of some of the factors responsible for tsetse transmission. Focus group discussions among healthcare workers aimed at understanding the adequacy of the healthcare delivery system in the district for the management of rHAT, in addition to seeking to identify challenges with respect to access to the healthcare system faced by rHAT patients, including economic and social challenges. In addition to the structured questionnaires, a key informant interview was undertaken with an rHAT specialist health worker to understand the case management of rHAT patients at the UTH, which is the highest referral centre. 

A questionnaire was administered to medical officers (District Health Management Officers) for the purpose of assessing the adequacy of the healthcare delivery system for the management of rHAT at the district level. All professional health workers who had worked at any of the health centres in the four districts and were responsible for disease detection were included in the study. The criteria for determining the qualified staff at the health centres for the purpose of this study were working as a healthcare practitioner (HCP) and responsible for disease detection. Those who had worked for less than one year at a health centre and had no experience handling rHAT cases were not eligible. Information collected included the number of health personnel available for the diagnosis of rHAT, staff knowledge of rHAT and the availability of drugs and equipment to diagnose and manage rHAT. 

We estimated the population at-risk using annual estimates for 2013 and 2014. According to the currently suggested methods for disease distribution and risk assessment at the national level, HAT elimination as a public health problem requires that fewer than 1 case/10,000 inhabitants/year (averaged over a five-year period) be reported from each health district [21]. A 2022 report was used to estimate the population at-risk on the basis of the number of HAT cases per annum. Risk was subsequently categorised as ‘very low’ (≥1 per 10^6^ people and <1 per 10^5^ people), low (≥1 per 10^5^ people and <1 per 10^4^ people), moderate (≥1 per 10^4^ people and <1 per 10^3^ people), high (≥1 per 10^3^ people and <1 per 10^2^ people) or ‘very high’ (≥1 per 10^2^ people). When a threshold of one HAT case per one million people per annum arises, the risk is considered “marginal”. Furthermore, when the risk categories are below the moderate level, i.e., low or very low, the objective of the WHO for the elimination of rHAT as a public health problem is met [21]. To determine the population at the district level, data from the Central Statistics Office Census of Population and Housing 2010 were used [22]. 

A Garmin version 10 geographical positioning system (GPS) receiver was used to collect waypoints to help determine the spatial distribution of all cases. An attempt was made to record the location of all the households with cases. However, poor road networks in the four districts and challenges in locating the households meant that not all locations could be recorded. The spatial resolution of the presented maps is relatively coarse so that households cannot be identified, and some locations overlay one another. Names of patients were not recorded, and strict confidentiality was adhered to throughout the study. Descriptive statistics were generated for all variables, and mapping was undertaken using ArcMap GIS version 10 (Esri, California United States of America). Shape files were obtained from the Zambia Statistics Agency (Zamstat), which is licenced under a creative commons licence.

### 2.4. Data Analysis

All structured interview data were entered and analysed using Excel™. Qualitative data from focus group discussions were entered into Microsoft Word™ and analysed manually based on accepted methods of coding and memo writing of qualitative data [13,23,24]. All transcripts were deidentified, and personal identifiers were removed to protect individual patient anonymity. For quantitative data, descriptive statistics were generated. A Fisher’s exact test cross-tabulation analysis was conducted to assess the proportion of people able to identify tsetse as the vector, the proportion aware of the mode of rHAT transmission, the proportion with knowledge of rHAT and the proportion that had encountered an rHAT case before being diagnosed. 

## 3. Results

### 3.1. Number of rHAT Cases 2004 to 2014

In total, 64 rHAT cases were included in the study (Table 1). Of these, 22 patients came from Chama, 11 from Rufunsa, 28 from Mpika and 3 from Mambwe. All cases (*n* = 64) were traced, and all were interviewed during, at most, two visits without the loss of any cases. A total 26 were identified through active surveillance, and were identified through 38 passive surveillance [13].

### 3.2. Knowledge of rHAT Transmission

The overall number of people able to identify tsetse as the vector and aware of rHAT transmission was 57 (*n* = 64); there was a significant association between identifying the vector for rHAT and knowledge of rHAT transmission according to Fisher’s exact test (one-tailed, *p* < 0.028). The analysis also indicated (*n* = 64) that there was no association between knowledge of rHAT and previous experience with an rHAT (two-tailed, *p* > 0.233). 

### 3.3. Common Signs and Symptoms Developed by rHAT Patients

The common signs and symptoms of rHAT experienced by patients were determined and are presented in Figure 1. The most common signs and symptoms among rHAT patients (with one missing value in the data) were fever (59 cases of 63), body pain (58 cases of 63), sleeping disorders (41 cases of 63) and headache (58 cases of 63). The abovementioned symptoms were reported in both male and female cases. Approximately 77% (35 of 45) of male cases were detected in stage 2 (the neurological stage of rHAT), with at least seven mortalities. A proportion of 37% (9 of 19) of female cases were detected in stage 2 of the disease, with one mortality.

### 3.4. Case Management of rHAT Patients 

The rural health centres in Mambwe (Masumba, Kakumbi and Nsefu), Rufunsa (Lukwipa and Shikabeta), Mpika (Nabwalya) and Chama (Kamufupu) districts did not have laboratory staff able to detect rHAT. The referral centres for Mambwe (Kamoto Mission Hospital), Rufunsa (St. Luke Mission Hospital) and Chama (District Health Hospital) had microscopes for disease diagnosis. None of the referral centres and rural health centre staff had attended refresher courses for rHAT detection. None of the referral centres had drugs for the treatment of the disease at the time of the visit. This could have been because the rural health centres did not have pharmacies but only dispensaries. 

Key respondents described the case management and common challenges associated diagnosis and low index of suspicion. It was highlighted that standard protocols for rHAT detection are used in all cases, and the prescribed drugs for treatment are suramin in the first stage and melarsoprol in the second stage: 


*The standard protocol for detection of rHAT is used on all cases received at UTH. This involves screening for potential infection, diagnosing by establishing whether the parasite is present in the body fluids, and staging to determine the level of the disease’s progression. The second stage disease detection involves a lumbar puncture. The drug used for treatment of the first stage patient is suramin and in the second stage melarsoprol.*


The key respondents also explained some challenges associated with treatment of rHAT: 


*A common challenge in the treatment of sleeping sickness is organ failure. A case was recorded of a 49 year old game ranger who was admitted to UTH hospital. His lung was normal and there was no leg swelling. Early stage HAT was detected by both microscopy and SRA-LAMP. Another example related to delays in diagnosis: A female adult aged 58 was admitted with fever and a reduced level of consciousness. On examination she was noted with lethargy and had an abnormal lung. However, late stage HAT was confirmed by microscopy and SRA-LAMP which was missed for several weeks while being treated for other conditions. Due to the delay in diagnosis, she died eight days after starting melarsoprol treatment. Not all HAT cases present the same signs and symptoms. This depends also on the index of suspicion from the onset of diagnosis.” Although it is recommended that patients come for review after completing the treatment, the key respondents indicated that this rarely happens at UTH: “It is also recommended that patients come for review at UTH. However, we have no record of a patient who came for review after being discharged. For example, we had a case of re-occurrence of infection in Rufunsa (St Luke’s hospital). This was after the patient was discharged from UTH. However, there has been no communication regarding the patients who are re-admitted at the various districts.*


### 3.5. The Population at Risk and Land Cover

The population at risk of rHAT cases and the associated land cover area in the four districts are described in Table 2. Chama and Mambwe districts had a low risk, whereas Rufunsa and Mpika Districts had moderate risk. Mpika had the highest rural and urban populations, as well as land cover, compared to the other three districts. 

### 3.6. Spatial Distribution of rHAT Cases in Each District

The spatial distribution of households with rHAT cases in the four districts is shown in Figure 2, Figure 3, Figure 4 and Figure 5. Thirty-eight cases were mapped both through active and passive surveillance. A total of 18 waypoints were selected, which could not represent all 38 cases individually due to overlap on the maps. The figures show that all areas with reported rHAT cases were located near or in game management areas. In Chama District (Eastern Province), the spatial distribution of the 22 rHAT cases was focused around two major points: one close to the border with Malawi and the other around Chama township (Figure 2). 

The cases of rHAT patients around the township occurred mostly in businessmen and women but included some teachers, who reported having crossed the game management areas at some point during their duties in the weeks preceding the onset of the disease. Teachers reported having crossed the game management area when they were assigned to attend to student examination in Mpika District, and businessmen (or women) reported crossing the game management area during the time they sold merchandise to communities in the rural parts of the district. In Mambwe District, all three reported cases of rHAT were concentrated one area, all occurring in fishermen working in the nearby river (Figure 3).

In Mpika District, the 28 rHAT cases were concentrated in two main areas within the Chiefdom of Nabwalya: (i) Kazembe and Uzimbwa villages and (ii) Dombo Department of National Parks and Wildlife Camp (Figure 4).

In Rufunsa District, the 11 rHAT cases were distributed mainly in three areas: (i) Shikabeta, (ii) Chomba and (iii) Lukwipa villages. These villages were all situated in game management areas (Figure 5).

During the focus group discussions, community members observed that the number of tsetse flies had been increasing across all four districts, and they felt that there were no clear mechanisms to control transmission: 


*The transmission of rHAT is also due to the game animals especially elephants and Buffalos. Encroachment of the human habitable land is very common. It’s not always the people who go in the game reserve areas. As a result, they come along with the tsetse flies and shed them in the communities. Currently, as residents we don’t see much of the mechanisms available to control the game animals they are elusive.*


The community members felt that several measures need to be undertaken to manage transmission. For instance, some members felt that tsetse fly control centres should be reintroduced for effective control and monitoring. Others felt that there should be a localised means to control game animals to avoid unwanted movements, and others explained that households near and within game management areas (e.g., homes of wildlife officers) should be supplied with effective tsetse repellents. Some community members also observed that the sprays (target) used against tsetse flies did not provide the necessary protection because children in the game management areas complained of tsetse bites, and they felt that these were more effective against mosquito bites.

### 3.7. Healthcare Delivery Systems

A total of 11 health facilities were covered in the study: six rural health centres and five hospitals, as shown in Table 3 and Table 4. All five district hospitals had a simple microscope (Table 3). Only one health centre (in Mambwe District) had a qualified laboratory technician. Rufunsa District had the largest number of staff able to diagnose rHAT (seven) compared to hospitals in other districts that had previously diagnosed an rHAT case; followed by Chama (four), Mpika (three) and Mambwe (two) districts.

Systematic documentation was provided for the rHAT cases recorded in Rufunsa District from 2012 onward. The Rufunsa area was under the jurisdiction of Chongwe District prior to 2012, when it was designated as a district in its own right. Hence, there are no separate rHAT records for Rufunsa District before 2012. Health staff in Chama District also diagnosed rHAT cases, although no records of active surveillance by health staff were found for the disease (Table 4). In Mpika District, most rHAT cases were diagnosed at Chilonga Mission Hospital, with referrals mainly from Nabwalya Rural Health Centre, a facility with no diagnostic equipment and no qualified professionals (Table 4).

None of the health centres visited in any of the study districts had any drugs for the treatment of rHAT. The process for requesting drugs was as follows: once a case was detected, a request was sent to UTH in Lusaka, followed by a request to the Ministry of Health, which, in turn, made a request to the WHO country office. The delivery period for the requested drugs was approximately 2−7 days, depending on transportation availability, which, at times, was not readily available. 

## 4. Discussion

In this study, we examined the spatial distribution of cases of rHAT from 2004 to 2014 in four study districts in Zambia and investigated the adequacy of the healthcare system to deal with rHAT. We selected districts in which sporadic cases of rHAT were reported [13]. The evidence shows that the availability of services and the capacity of health facilities to diagnose rHAT in these districts was limited.

The number of rHAT cases included in this study was *n* = 64. The low prevalence arises primarily because rHAT is a zoonosis that does not often spill over to the human host. Moreover, the number of cases is likely to be underestimated [2] due to under-reporting, misdiagnosis, poor health-seeking behaviour and inactive surveillance (no reported physical activity) in the four districts [13]. Despite the low prevalence, analysis of the 64 cases has the potential to identify any gaps in the Zambian national healthcare system, which is critical to achieving the goal of eliminating sleeping sickness.

Most of the respondents (94%) across the gender, district and occupation categories had a general knowledge of rHAT transmission and the vector responsible for its transmission. The findings are similar to those reported by Bukachi et al. [24], who found that communities in the tsetse-infested areas of South Sudan had 90% general knowledge of HAT. The general knowledge of people in the four districts investigated in the present (Chama, Mambwe, Rufunsa and Mpika) could act as key contributor to the elimination of rHAT as a public health problem, as targeted by the WHO [25]. Knowledge about rHAT transmission among communities should be considered an element that contributes to the two primary indicators used to measure progress toward the elimination of rHAT as a public health problem, which are sustained active surveillance and improved reporting systems [21]. Once sufficient knowledge is available among members of the community, poor health-seeking behaviour and misdiagnosis are more likely to be avoided, resulting in more accurate statistics on the true number of cases [13]. Existing knowledge and awareness in communities should be blended with continued education about the disease, with the aim of elimination [26]. This can be achieved by a One Health approach to prioritisation of African trypanosomiasis as an important neglected disease in Zambia and for the formulation of One Health strategies for improved control in communities [27]. A One Health approach may achieve cost-effective, sustainable rHAT surveillance and control by strengthening health systems [28].

In Chama District in Eastern Province, most rHAT patients from one village (Munyakanyaka village), crossed the Zambian border and sought treatment at Lumpi Mission Hospital, a referral centre in Malawi, which was outside our study area. The patients crossed the Zambian border because Chama District Hospital did not stock the drugs required to treat rHAT. In most cases, rHAT patients were referred to UTH, which is the largest tertiary hospital in Zambia. However, patients preferred to seek health services at the Malawian hospital, as it was much closer and more accessible to the local people in Chama District. Although Mambwe District has a well-equipped laboratory with one microscope and four laboratory specialists, no deliberate active rHAT surveillance took place in the district. Consequently, the true incidence of cases of the disease in the district was not available. In Mpika District, Chilonga Mission Hospital was found to be the only hospital and referral centre admitting rHAT patients for treatment. Antitrypanosomal drugs were administered by medical doctors, medical licentiates, clinical officers and trained nurses in this hospital. 

The referral centre for Rufunsa District was St. Luke’s Mission Hospital. Although rHAT cases were still referred to UTH, St. Luke’s Mission Hospital was able to handle rHAT cases emerging in the district. Antitrypanosomal drugs were available only at the district hospital upon request from UTH, which also had to make a request to the Ministry of Health. Disease-endemic countries, such as Zambia, are provided with drugs according to forecasts of usage. In non-disease-endemic countries, pharmacy services in hospitals diagnosing and treating rHAT must address requests for drugs to the WHO [25,26]. The conditions for requesting drugs should also be accompanied by epidemiological and clinical data on the patient, as well as contact details of the hospital and the medical doctor in charge of treatment. The WHO ensures delivery of drugs within 24 to 48 h [25]. However, delays are commonly incurred in transporting the drug to remote rural areas where rHAT cases are often reported. 

We found that the Zambian healthcare delivery system was ill-equipped to handle rHAT cases, a situation also highlighted by Mulenga et al. [29]. Standard laboratory equipment was not readily available in most of the studied hospitals and clinics, which could be attributed to low and inconsistent allocation of resources to remote areas. Kunda et al. [30] also reported that laboratories, particularly in the rural areas where rHAT cases are often identified, were poorly equipped and were not able to diagnose many emerging and re-emerging diseases. Lejon et al. [31] reported similar findings with respect to gHAT, although more diagnostic options existed than for rHAT; this demonstrates that integration of HAT control into the health system is hindered by many factors, including the limitations of current diagnostic tests. Besides this lack of resources, awareness and training, the motivation to test for such diseases amongst health workers and the community was also found to be an important contributor; therefore, there efforts should be made to improve understanding. 

Opportunities for refresher courses on zoonotic diseases were not provided to health staff in the studied areas, meaning that awareness and levels of competence remained low among health practitioners who manage rHAT (and other zoonotic diseases). Kunda et al. [30] similarly reported that some zoonoses were likely missed by those entrusted with the duty of identifying them. Lack of awareness is widespread among practitioners, who were more likely to concentrate on more commonly diagnosed diseases in their catchment area at the expense of diseases such as rHAT, despite their public health importance [32]. Record keeping in all the facilities visited was also unsatisfactory, contributing to inaccurate estimation of the number of diagnosed rHAT cases and underestimation of the true (i.e., diagnosed and undiagnosed cases) incidence in the study area. 

The weakness of the public health system has cost implications for affected households, such as lost income when a person is seeking treatment, which increases the burden of the disease (beyond the direct effects of morbidity and mortality) on the household. According to Shaw et al. [32], decision making and financial planning for HAT and tsetse control are complex in that they require a particular range of choices to be made, such as those relating to timing, methods, strategies and finance. With the WHO target of eliminating HAT likely to be achievable (without correcting for under-reporting of rHAT), a holistic approach to improving health service delivery will be needed if this is to be sustainable through the next decade [27].

To estimate the population at risk, we used annual estimates for 2013 and 2014. The period between 2013 and 2014 was a time of active surveillance of rHAT cases in the four investigated districts (Mambwe, Mpika, Chama and Rufunsa). Previous studies estimated sleeping sickness at the continental, regional and national levels based on the educated guesses and rough estimates of experts rather than using epidemiological evidence of the population at risk [21]. Our study reveals that Rufunsa and Mpika were at moderate risk, whereas Chama and Mambwe districts were at low risk, similar to the results reported by Franco et al. [21], who analysed a five-year period. The low risk recorded for Chama and Mambwe districts can likely be attributed to under-reporting. The results show that community members in Chama District sought medical care from in neighbouring country of Malawi, particularly Lumpi Hospital.

The epidemiological distribution of rHAT is affected by the presence of wild and domestic animals, such as livestock and dogs, which can act as disease reservoirs [33,34]. As the reservoir community for rHAT in Zambia includes a broad range of wildlife species, proximity to wildlife and tsetse habitats can be risk factors for transmission [35]. In our study, more than 90% cases were reported in areas that close to game management areas, which are tourist attractions and foci for rHAT transmission [33]. As cases can also occur in tourists [32], rHAT has the potential to reduce tourism in game management areas if not controlled and indirectly negatively affect the economy of Zambia. The WHO [25] reported that land-use pressure has resulted in an increasing overlap of the grazing areas for wildlife and domestic animals. This poses a threat of increased transmission of human infective trypanosomes from wildlife to humans or to their livestock [36].

Early and accurate diagnosis of rHAT is critical for effective management of the disease [37]. However, this is difficult to achieve in remote rural areas where the disease is endemic for several reasons. First, the clinical signs of rHAT are similar and usually confused with those of more common endemic febrile diseases, such as malaria, tuberculosis and HIV/AIDS. Secondly, the inadequacy of the microscopes in the rural health centres, which can detect malaria (*Plasmodium falciparum*) and rHAT using stained blood film microscopy, has caused considerable dependency on rapid diagnostic tests for malaria detection. The use of these tests for malaria detection means that blood smears are less commonly examined, and the likelihood of diagnosing rHAT is reduced. The use of the microscopic stained blood film method for malaria has additional benefits; it is cheap and enables the person performing the diagnosis to score parasite density, identify the different *Plasmodium* species, differentiate sexual (gametocytes) from asexual stages and avoid accidental detection of other haemoparasites, such as trypanosomes in the blood smear [38]. As such, the use of microscopy in such resource-limited rHAT and malaria-endemic regions, including regular training of health personnel involved in disease diagnosis, is crucial. Although microscopy is associated with low sensitivity due to fluctuating parasitaemia in gHAT patients, parasitological confirmation is easier in rHAT patients because bloodstream trypanosomes are numerous [20,33]. On the other hand, the requirement for specific equipment, reagents and highly skilled human power in some cases, can limit the application of molecular tests in such remote and resource-limited areas. 

Currently, implementing effective case management algorithms in the study areas is challenging [39]. Due to misdiagnosis, the majority of rHAT cases are reported late in the second stage of the disease, resulting in organ failure and treatment complications. Furthermore, it is possible that these cases may be indicative of others that are unreported. According to Odiit et al. [40] about 39% of rHAT cases and 92% of rHAT are unreported. To develop improved rHAT management algorithms, we propose the following: (i)Enhancement of rHAT (and its transmission dynamics) advocacy and sensitisation among both health personnel and local communities in tsetse-inhabited rural areas. This is expected to increase the suspicion index among health workers and promote earlier reporting of suspicious cases to health centres by local communities;(ii)Governments in tsetse-inhabited regions should be urged to invest in basic diagnostic equipment, such as microscopes for detection of various haemoparasites, including plasmodium and trypanosome parasites, as well as detection of ectoparasites and endoparasites. Each rural health centre should have at least one microscope and a trained technician, who should attend refresher courses regularly. Both (i) and (ii) are ingredients of early and accurate detection of rHAT cases, which should lead eventually to successful case management with minimal complications;(iii)In referral centres with advanced diagnostic capacity, regular screening of domestic animals, wildlife and tsetse flies for trypanosome species by means of microscopy and more sensitive molecular tests such as PCR and, in particular, the user-friendly and cost-effective LAMP, should be encouraged;(iv)Because wild animals are natural reservoirs of *T. b. rhodesiense*, local communities and their domestic animals should be discouraged from encroaching into game management areas and national parks in order to minimise interactions with wildlife and tsetse flies. Furthermore, local communities should be sensitised to the dangers of poaching and related activities, which expose them to tsetse bites, and the management of rHAT should be considered when developing management plans for game management areas;(v)Health workers in rural health centres should be trained in basic record keeping and its significance in management and control of diseases, including rHAT;(vi)Governments in tsetse-inhabited regions should be encouraged to invest in effective tsetse control measures, such as insecticide-impregnated targets or aerial spraying with pyrethroids. Considering that rHAT is a transboundary disease, countries sharing boundaries in such regions should work closely and conduct such activities jointly to more sustainably control tsetse flies and subsequently rHAT.

Primary healthcare systems face challenges in the treatment of rHAT, partially due to inadequacies of the current treatment options, which are considered scarce, toxic, ineffective and difficult to administer for stage 2 of rHAT. However, the development of a new treatment called fexinidazole, a 2-substituted 5-nitroimidazole rediscovered by the Drugs for Neglected Diseases initiative (DNDi), could provide significant help in these rural areas, where there are no skilled human resources or pharmacies to stock and dispense drugs [41]. Fexinidazole could provide a short-course, safe and effective oral treatment to cure both acute (stage 1) and chronic (stage 2) rHAT infection. The drug has the advantage that it can be administered at the primary healthcare level without supervision of skilled human resources [41]. The availability of the drug would also reduce health consequences, such as pain, amnesia, physical disability, stigma, dropping out of education and loss of friends and self-esteem, as reported by Mwiinde et al. [13]. 

Due to the scarcity of resources in Africa, especially in rural remote areas, where the disease is often found, such as in Zambia, the challenges of disease diagnosis are significant. As such, some infected individuals may die before they can be diagnosed and treated [41]. Early detection of rHAT by determining its presence in the body fluids should be carried out in rural health centres. However, staging to determine the state of disease progression only needs to be effectively incorporated at referral centres and UTHs using skilled human resources due to the need for clinical examination and, in some cases, analysis of the cerebrospinal fluid obtained by lumbar puncture, a complicated procedure that carries risks [42].

The geographical distribution of rHAT is expected to be sensitive to human population growth and naïve animals [43]. Moreover, climate change can alter the relationship between humans, tsetse flies and the environment, thereby altering the likely spatial distribution of rHAT, which itself is sensitive to these factors [44,45]. Furthermore, major environmental changes are likely to occur in parts of Africa in the coming decades, which may alter the spatial distribution of rHAT. Therefore, policies should be formulated in the sub-Saharan region to plan for a changing risk landscape [44,45]. Important in this context is the cross-border collaboration of Zambia and its neighbouring countries, particularly Malawi, where the border region has a high prevalence of rHAT with two naïve populations. In this study, we were not able to investigate the host genetic differences between the geographical locations, but this could be investigated in future research. 

Under-reporting and poor data management of rHAT cases in the four investigated districts, as highlighted by Fèvre et al. [2] and Mwiinde et al. [13], made it difficult to determine the geographical distribution of rHAT risk. 

## 5. Conclusions

The healthcare delivery system in Zambia was found to be ill-equipped to handle rHAT cases in the studied districts in terms of both physical resources, such as laboratories, and human resources, including training. There is an urgent need to improve rHAT advocacy and sensitisation, disease detection and management, and policies regarding human resource management. There is also a need for continuous surveillance of *T. b. rhodesiense* in humans and domestic and wild animals, as well as the tsetse vector in tsetse-infested areas, particularly in areas reporting rHAT cases. A One Health approach may help to combine the necessary intervention programmes aimed at eliminating rHAT as a public health problem. We conclude that improvement in detection, recording/reporting and access to curative drugs is necessary in Zambia.

## Figures and Tables

**Figure 1 tropicalmed-07-00270-f001:**
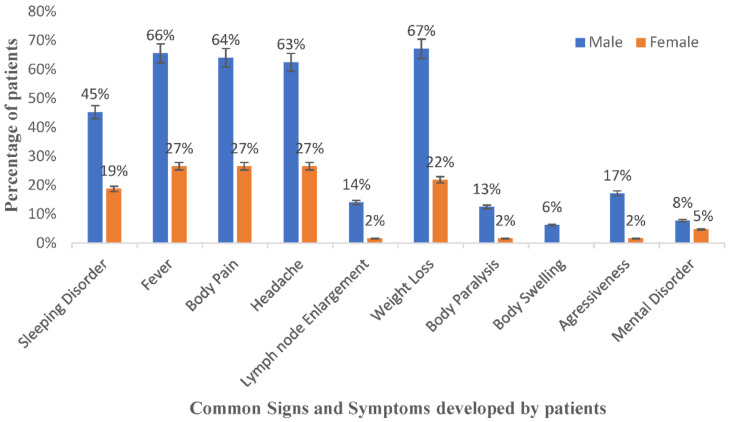
Signs and symptoms developed at onset of rHAT by gender.

**Figure 2 tropicalmed-07-00270-f002:**
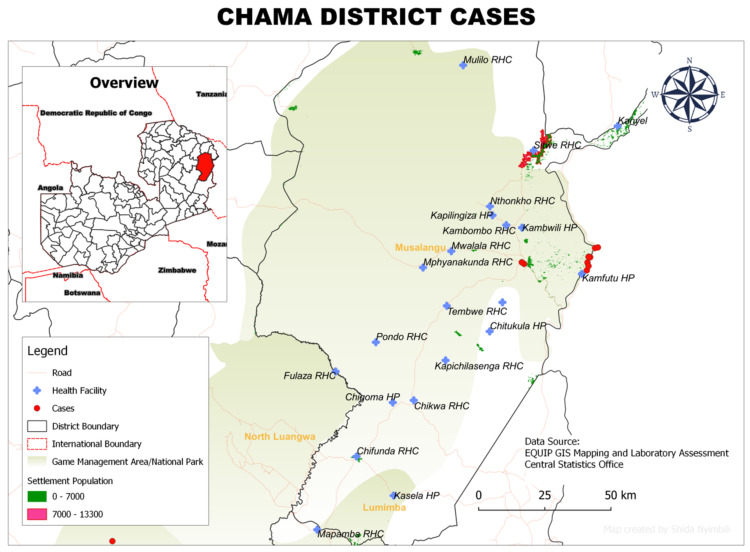
Spatial distribution of rHAT cases from 2004 to 2014 in Chama District, Muchinga Province, Zambia.

**Figure 3 tropicalmed-07-00270-f003:**
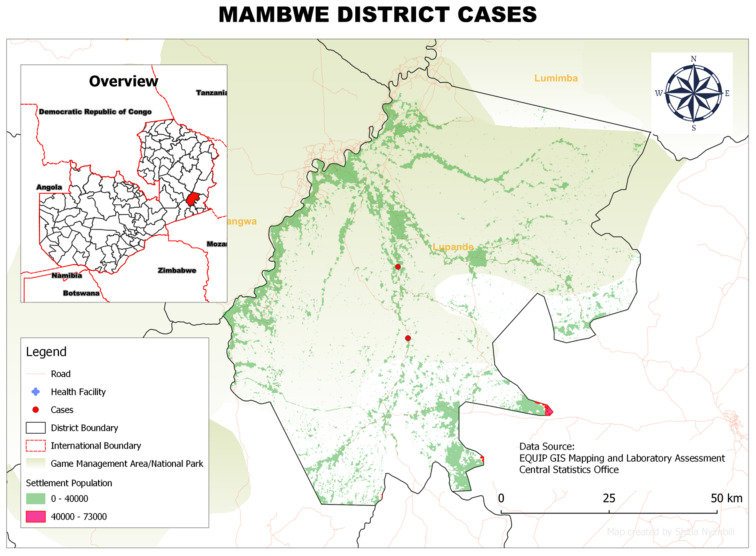
Spatial distribution of rHAT cases from 2004 to 2014 in Mambwe District, Eastern Province, Zambia.

**Figure 4 tropicalmed-07-00270-f004:**
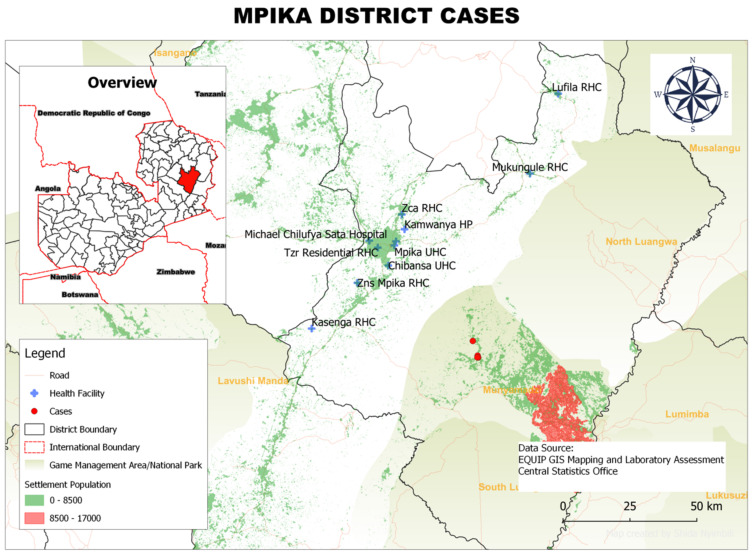
Spatial distribution of HAT cases from 2004 to 2014 in Mpika District, Muchinga Province, Zambia.

**Figure 5 tropicalmed-07-00270-f005:**
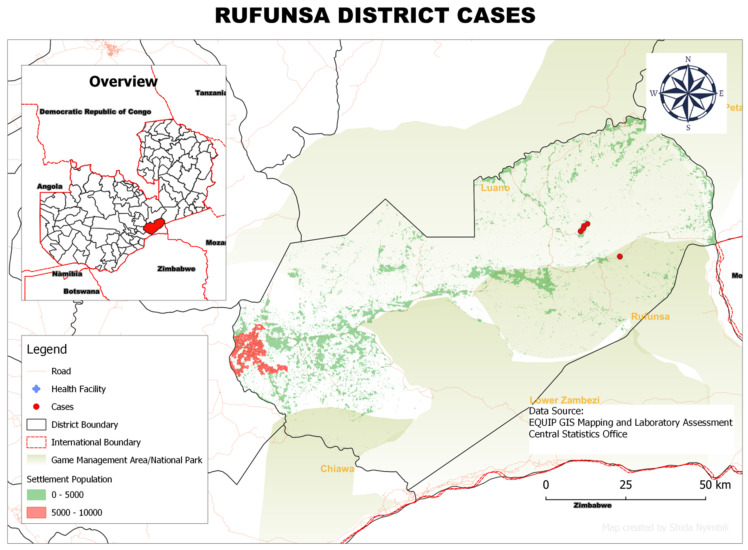
Spatial distribution of rHAT cases from 2004 to 2014 in Rufunsa District, Lusaka Province, Zambia.

**Table 1 tropicalmed-07-00270-t001:** The total number of HAT cases diagnosed from 2004 to 2014 across the four districts studied.

Year	Male	Female	Chama	Mambwe	Mpika	Rufunsa	Total
2004	3	0	3	0	0	0	3
2005	2	0	2	0	0	0	2
2006	4	0	3	0	1	0	4
2007	2	0	2	0	0	0	2
2008	3	2	3	1	1	0	5
2009	2	0	0	1	0	1	2
2010	3	1	4		0	0	4
2011	1	1	0	0	1	1	2
2012	3	0	1	0	0	2	3
2013	9	5	1	1	8	4	14
2014	13	10	3	0	17	3	23
Total	45	19	22	3	28	11	64

**Table 2 tropicalmed-07-00270-t002:** The population at risk of contracting rHAT and associated land cover area.

District	No. of Cases in 2014	No. of Cases in 2013	Total No. of Cases in 2013 and 2014	Rural Population at Risk	Total District Population at Risk	Land Cover (km^2^)
Chama	3	1	4	92,620	99,434	17,630
Rufunsa	3	4	7	71,000	71,000	9614
Mpika	17	8	25	154,199	191,329	40,935
Mambwe	2	1	3	59,076	64,627	5294
Total	23	14	37	376,895	649,946	73,477

**Table 3 tropicalmed-07-00270-t003:** Adequacy of the healthcare delivery system for the management of rHAT in Mambwe and Rufunsa districts.

	Mambwe District				Rufunsa District		
Health centre	Kamoto M/H	Masumba	Kakumbi HC	Nsefu RHC	St Luke M/H	Lukwipa RHC	Shikabeta Rural H/C
Referral centre	UTH	Kamoto M/H	Kamoto M/H	Kamoto M/H	St Luke M/H	St Luke M/H	St Luke M/H
Equipment	Microsc *	0	1	0	Microsc *	0	0
Number of qualified staff	5	3	4	2	9	1	1
Number of staff able to diagnose HAT	4	0	3	0	7	1	1
Number of laboratory staff	2	0	2	0	3	0	0
Laboratory staff refresher course available?	No	No	No	No	No	No	No
Number of HAT cases Encountered	1	0	1	0	10	1	0
Pharmacy?	Yes	No	Yes	No	Yes	No	No
Drugs available?	No	No	No	No	No	Yes	No

UTH: university teaching hospital, M/H: mission hospital, HC: health centre, RHC: rural health centre, Microsc *: simple microscope. Number of staff refers to those able to diagnose rHAT through both laboratory and clinical approaches.

**Table 4 tropicalmed-07-00270-t004:** Adequacy of the health delivery system for the management of HAT in Chama and Mpika districts.

	Chama District		Mpika District	
Health centre	Chama DH	Kamfupu RHC	Chilonga	Nabwalya RHC
Referral centre	UTH	Chama DHMT	Chilonga M/H	Chilonga M/H
Equipment	Microsc *	0	Microsc *	0
Number of qualified staff	8	1	9	2
Number of staff able to diagnose HAT	6	0	4	0
Number of laboratory staff	2	0	2	0
Laboratory staff refresher course available?	No	No	No	No
Number of HAT cases encountered	<10	<10	<10	0
Pharmacy?	Yes	No	Yes	No
Drugs available?	No	No	No	No

UTH: university teaching hospital, M/H: mission hospital, HC: health centre, RHC: rural health centre, Microsc *: simple microscope.

## Data Availability

Publicly available datasets were analyzed in this study. This data can be found here in The Environmental Information Data Centre repository, available at http://doi.org/10.5285/6f70d562-8fcf-4ecd-adaf-cbc5800cc326 (accessed date 7 July 2021).

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
