# Peer review of "Healthcare Management of Human African Trypanosomiasis Cases in the Eastern, Muchinga and Lusaka Provinces of Zambia"

_tropicalmed, 2022, doi:10.3390/tropicalmed7100270_

Round 1

Reviewer 1 Report (Previous Reviewer 2)

Line 44 and 45 X2(1): The (1) is which represents the degrees of freedom (df) used to calculate your chi square results. I suggest it must be removed 

Secondly, if the chi square results are based on k=1, then your P values and chi square results are calculated incorrectly  

Line 263: Mwiinde et al,** (insert reference number for this citation)

Lines 285, 287 and 288 write the province with a capital letter P

Section 3.2 Knowledge of rHAT transmission: Please recalculate the chi square and p values

Figure 1: Inset error bars on the graph to accommodate the sampling bias used on the uneven number of samples 

Line 790: In Chama district (Eastern province): Write district and province as District and Province. The same for Figure 2 caption as well as lines 857 and 858

Please do the same for Figure 3 caption and line 869

The same for Figure 4 caption and line 877, 902, 903 and 1363

Author Response

Attached is the point-by-point response to the reviewer number 1.

Reviewer 2 Report (New Reviewer)

In this manuscript, Mwiinde et al describe human African trypanosomiasis cases in Zambia, and their relationship to patient knowledge and availability of diagnosis and treatment. Overall, this is in interesting and appropriately designed manuscript. However, I have the following major concerns:

1.     Figure 2-5 show different shadings of green. The interpretation of the different green colors should be clarified in the legend.

2.     Criteria used to define “qualified staff” and “staff able to diagnose HAT” should be provided.

Minor concerns:

1.     Figure 1 would be more effective if expressed in percentage. As-displayed, most disease manifestations appear to be more common in men, but this may be due to the greater number of male patients.

2.     Figure 1 x axis: lymphnode should read “lymph node”

3.     Line 253, lumber should read lumbar

4.     Line 319, “the spray’s” should read “the sprays” (no apostrophe)

5.     Line 326, “microscope (Tables 3).” should read “microscope (Table 3).”

6.     Line 557, rHAT abbreviation is not defined

Author Response

Attached is the point-by-point response to reviewer number 2.

Reviewer 3 Report (New Reviewer)

The article intituled “Health Care Management of Human African Trypanosomiasis Cases in the Eastern, Muchinga and Lusaka Provinces of Zambia", shows all efforts towards the number of infection cases that occurs by Rhodesian form of Human African trypanosomiasis btained from rural health facilities in four districts in Eastern, Muchinga and Lusaka provinces of Zambia. The subject presented by authors has a great interest to tropical medicine area. I consider it a well-done study that was interesting explored and conduced. Thus, I suggest that the article undergo a minor review to improve it.

-Minor changes:

1) Adjust all references, [1] until [49], according to the journal's rules, as they are all very different from each other (examples below):

“6. Bonnet J, Boudot C, Courtioux B Overview of the Diagnostic Methods Used in the Field for Human African Trypanosomi-1881 asis: What Could Change in the Next Years? Hindawi Publishing Corporation (2015): 10:1155. 1882

7. Franco R.J, Cecchi G, Priotto G, Paone M, Diarra A, Grout L, Simarro P.P, et al., Monitoring the elimination of human 1883 African trypanosomiasis at continental and country level: Update to 2018, PLoS Negl Trop Dis 14(5): (2020).”

2) Please, double check in reference [29].

3) Add “rHAT” in abbreviations section.

4) It's so confusing, I suggest deleting this section:

 “6. Patents

This section is not mandatory but may be added if there are patents resulting from the work reported in this manuscript.”

3) Please, authors must leave the abstract section in the journal format (rules below):

“The abstract should be a total of about 200 words maximum. The abstract should be a single paragraph and should follow the style of structured abstracts, but without headings…” (https://www.mdpi.com/journal/tropicalmed/instructions).

Author Response

Attached is a point-by-point response to reviewer number 3. 

Reviewer 4 Report (New Reviewer)

This is very well written manuscript that will provide a reference point for HAT researchers. The goals are well stated and the authors data confirms the need for significant improvements in detection, reporting and treatment regimens of HAT in this geographical area. 

A minor comment which should not detract from the objective of the study is that it may be helpful to the reader to show the prevalence (N) of gHAT and rHAT based on the total number of cases at the clinic, if this data is available. It would certainly be possible to statistically analyze the data for each site in terms of N for gHAT and rHAT.   

Author Response

Attached is the point-by-point response for the reviewer number 4

This manuscript is a resubmission of an earlier submission. The following is a list of the peer review reports and author responses from that submission.

Round 1

Author Response

We wish to thank the reviewer for the guidance provided to improve the manuscript. 

Reviewer 2 Report

Abstract

Line 24: Delete second spacing between 2014 and in

Line 25: Delete full stop between and. Case and make C from case lower caps

Line 43: Write T. b. rhodesiense in italics

Line 52: Put spaces between T.b. and make the G in Gambiense small caps

Line 56 and Line 59: Put spaces between T.b 

Line 56 and Line 59: Write T. in full as you can not a sentence with an abbreviation 

Line 92: Put space between location of

Line 106 and 107: Put a line space section 2.1 and section 2.2 

Line 156: Delete second full stop on social challenges

Line 175 until 188: Inconsistent paragraph spacing. Please correct

Line 198: Put space between section 2.4 and the last paragraph of section 2.3

Line 210: Inconsistent sub heading. 3.1 must be in italics as the previous subheadings 

Line 212 and 213: Delete inverted commas 

Line 215: passive surveillance13? What is the 13 for? Please correct 

Line 215 and 216: Insert space between table 1 and the last paragraph of section 3.1

Table 1 can be improved by conducting chi square analysis to determine the significance in infection rate between males and females over the sampled decade 

Line 222 and Line 228: Chi square analysis is conducted however there is no mention of this in the materials and methods under Data analysis. Please revise and correct

Line 219, Section 3.2: I suggest you also summarize your results in percentages 

Line 226: Write 57 in words as you can not a sentence with a number 

Figure 1: I suggest you include standard error bars as the numbers of samples used are uneven. The error bars will be to accommodate the error margin in the sampling bias 

Line 251: Delete repeated words "rural health centres 

Section 3.4: Inconsistent paragraph spacing

Line 278 and Line 279: Put spaces between the last paragraph of section 3.4 and sub heading of section 2.5 

Table 2: Align your table columns with the heading columns 

Line 291: Write 38 in full as you can not start a sentence with a number

Figures 2 to Figure 5: The figure captions must be separated from the text. Please correct. Figure captions are not aligned with the text. Please correct

Table 3 heading is too far from the actual table. Please reduce the spaces between the heading and the table

Table 4: Please move "UTH: University Teaching Hospital, M/H: Mission Hospital, 411 HC: Health Centre, RHC: Rural Health Centre, Microsc*: Simple Microscope" to the bottom of the table like in table 3 and appropriate use of footnotes in both tables or use a different font size 

4. Discussion: Inconsistent paragraph spacing. Please correct 

Line 545: write AND in lower cases 

Line 594: Write Plasmodium in italics 

Line 630: Write GMAs in full as it is the 1st time it is mentioned 

Line 650 and Line 676: Remove the citation years  as it is inconsistent with the previous citations 

Line 671: (Figure 3). Please correct to Fig.3 as in the other figures 

Reference number 7, 8, 37, 39, 42, 46 and 49 are incomplete and missing some authors. Please correct  

Line 676: et al. is in italics and the rest in normal font. Please correct 

Line 679: WHO is reference number 53 and not 54. Please correct and remove reference 54 as your citations end at 53.  

In general: I suggest you revise the data analysis section to include all analysis conducted. 

For results, summarize in % and not in actual numbers found 

Author Response

(The authors gave the same response as above.)

Round 2

Reviewer 1 Report

General comments

Although I can see that a few of the issues raised in my first feedback have been addressed, the manuscript at this stage is not clear enough.

Based on the provided information, this paper provides an interesting overview of HAT patients’ characteristics, cases location, and an “insight” of the capacity of the health care system in managing HAT patients.

However, this paper still shows a mismatch and inconsistency between the initial objectives and what the authors have achieved and can conclude based on the data they have collected and the descriptive analysis they did, which may hinder some technical problem and methodological flaws.

Also, many parts of the introduction and discussion are not necessary and could be cut, and some interesting points such as the difference between number of cases in male and female cases could be discussed (although the difference may not be statistically different).

I would suggested to resubmit it after restructuring it and having the support of an English native speaker in editing it for language and style needed for a scientific publication.

Below some general suggestions, which I hope will be helpful for the authors.

Abstract

The abstract need substantial reshape. I would suggest perhaps to read online guidelines on how to structure an abstract. 

Also, there are several redundancies, grammar errors and language style that need to be addressed to obtain a clear and concise abstract.

Reshaping the article

The objectives of the authors were:

·         a) To map the geographical location of the HAT cases that have been detected between 2004 and 2014 in four districts

·        b)  To assess the population at risk

·         c) To assess environmental factors associated with transmission of the disease

·         d) To assess the adequacy of the health care delivery system for their management in the districts

The whole article, including the abstract, could be built in paragraphs related to these objectives of the manuscript:

1.       Mapping the geographical location of HAT cases

2.       Assessing the population at risk

3.       Assessing environmental factors related to HAT

4.       Assessing the adequacy of the health care delivery system

Starting from the methods, each sub-paragraph related to each of the above objectives, could clearly define:

a)       What data was collected to achieve that specific objective

b)      Where, when and how it was collected (retrospective analysis, patient data, interviews etc)

c)       How the analysis was done

Results could be written in the same way, e.g. with paragraphs with the headlines as above, and the results directly associated with each of the objectives.

The discussion should stick to the results and above all, the conclusions should be strictly linked to what the data describe (3-4 main key messages related to the results and objectives).

Finally, it may be useful to look at other publications that report qualitative data to see how this type of data is reported and other publications that describe health care system capacity and relationship with disease management (even if not HAT). It may help in structuring this paper.